# Sources of widefield fluorescence from the brain

Jack Waters*

Allen Institute for Brain Science, Seattle, United States

**Abstract** Widefield fluorescence microscopy is used to monitor the spiking of populations of neurons in the brain. Widefield fluorescence can originate from indicator molecules at all depths in cortex and the relative contributions from somata, dendrites, and axons are often unknown. Here, I simulate widefield illumination and fluorescence collection and determine the main sources of fluorescence for several GCaMP mouse lines. Scattering strongly affects illumination and collection. One consequence is that illumination intensity is greatest ~300–400 µm below the pia, not at the brain surface. Another is that fluorescence from a source deep in cortex may extend across a diameter of 3–4 mm at the brain surface, severely limiting lateral resolution. In many mouse lines, the volume of tissue contributing to fluorescence extends through the full depth of cortex and fluorescence at most surface locations is a weighted average across multiple cortical columns and often more than one cortical area.

## Introduction

Widefield fluorescence microscopy is a popular technique for monitoring activity in the mouse neo-cortex, often used in combination with genetically-encoded calcium indicators (*Mohajerani et al., 2013*; *Wekselblatt et al., 2016*; *Allen et al., 2017*; *Makino et al., 2017*; *Mitra et al., 2018*). A defining characteristic of widefield microscopes is a lack of optical sectioning. In some mouse lines, fluorescent indicator molecules are expressed at many depths in neocortex, either in multiple cell populations or throughout neurons that extend across cortical layers (*Daigle et al., 2018*). With no optical sectioning, fluorescence can and often does arise from multiple cortical layers and assignment of the signal to cellular or laminar populations is challenging (*Allen et al., 2017*). In addition, scattering by brain tissue commonly deflects photons en route to the camera chip, affecting lateral resolution (*Silasi et al., 2016*).

How severe are these problems? From which layers does fluorescence arise and how much does each layer contribute to total fluorescence? What volume of tissue contributes to fluorescence at a single camera pixel and what's the resulting resolution? Here, I answer these questions for several representative mouse lines, using a Monte Carlo random-walk model to simulate the propagation of photons in brain tissue.

Monte Carlo random-walk models provide an accurate and computationally tractable solution to the radiative transport equation and have been used extensively to simulate the propagation of photons through scattering media, such as biological tissues (*Zhu and Liu, 2013*). Random-walk simulations of near-infrared light propagation have been used to refine light intensity, source geometry and duty cycle in photodynamic therapies (*De Jode, 2000*; *Valentine et al., 2012*) and optimize illumination parameters and estimate the volume of tissue contributing to signals in diffuse reflectance tomography (*Boas et al., 2002*; *Fukui et al., 2003*). More recently, random-walk models have been used to explore brain illumination in the visible spectrum, often near the tips of optical fibers implanted to activate or silence neurons expressing opsins. Random-walk models can accurately predict the volume of tissue in which neurons are activated or silenced and have been used to refine stimulation parameters to minimize heating (*Bernstein et al., 2008*; *Kahn et al., 2011*; *Liu et al.,*

*For correspondence:
jackw@alleninstitute.org

**Competing interests:** The author declares that no competing interests exist.

*2015*; *Stujenske et al., 2015*; *Yona et al., 2016*). Fluorescence has been simulated by separating illumination and the fluorescence detection into two processes (*Chen et al., 2012*; *Holt et al., 2015*; *Hennig et al., 2016*) and widefield fluorescence has been studied for columnar arrangements of fluorophores, revealing that the effects of numerical aperture and focal position within the tissue depend on the size of the column (*Tian et al., 2011*).

Here, I adapted code previously used to model illumination by visible and near-IR illumination (*Stujenske et al., 2015*; *Podgorski and Ranganathan, 2016*; *Wang et al., 2020*), simulating illumination and fluorescence emission at visible wavelengths in a ~200 mm$^3$ volume of mouse grey matter. Spatially detailed models, containing individual tissue elements such as neurons and blood vessels, have proven invaluable for exploring optics with single-neuron resolution (*Charles et al., 2019*), but require substantial computational resources. I therefore modeled tissue as a spatially homogenous medium with wavelength-dependent scattering and absorption (*Wang et al., 1995*), a common simplification that requires fewer computational resources and has proven accurate where, as here, many photons propagate for multiple scattering lengths.

Widefield fluorescence begins with photons, usually from an incoherent source, propagating through the brain surface and into tissue. This illumination is attenuated by tissue absorption and scattering, and one might expect illumination intensity to decline with depth. Whether and at what depth illuminating photons are absorbed by fluorophore molecules depends on the illumination intensity and the density of fluorophores at different depths. After excitation, fluorescence photons propagate from the source molecule in random directions. Emitted fluorescence is subject to absorption and scattering before exiting the brain and being focused by the objective onto a detector, usually a camera. To understand how fluorophore molecules in different locations within the brain contribute to total fluorescence, I describe illumination intensity, fluorophore expression, and collection efficiency, and how these three factors change throughout the tissue for several mouse lines with GCaMP expression in laminar neuron sub-populations.

## Results

### Widefield illumination

I began by considering the spread of illuminating photons into brain tissue, simulating 480 nm illumination from an incoherent source focused into the brain through a cranial window by a low magnification, low numerical aperture objective, a configuration commonly used to image activity in mammalian brains using GFP-based indicators. As expected, the broad trend was of a decline in illumination intensity with tissue depth but with an increase in intensity over the initial 200 μm (*Figure 1A*). When the scattering coefficient was set to zero, eliminating scattering from the model, intensity declined exponentially from the tissue surface with a length constant matching that of absorption (*Figure 1A*). With the absorption coefficient set to zero, leaving scattering the only mechanism of attenuation, the superficial rise in intensity became larger and extended deeper into the tissue, indicating that the superficial increase in illumination intensity is a result of scattering (*Figure 1A*).

How does scattering increase the illumination intensity in superficial tissue? Photons exit the objective and enter the tissue propagating almost perpendicular to the brain surface (*Figure 1B*). The propagation angle is randomized over multiple scattering events. The cosine of the mean angle of propagation (relative to the optical axis; perpendicular to the tissue surface) relaxes towards zero over ~800 μm (*Figure 1C*), consistent with a calculated transport length of 826 μm. By randomizing propagation directions, scattering slows propagation perpendicular to the brain surface, concentrating photons.

The concentrating effect of scattering is lost when angles at the brain surface are randomized, such as by overlying skull. In mouse, widefield imaging is often performed through intact skull (*Mohajerani et al., 2013*; *Silasi et al., 2016*; *Allen et al., 2017*; *Makino et al., 2017*; *Gilad and Helmchen, 2020*; *Valley et al., 2020*). Mouse skull is ~150–300 μm thick and transparent but strongly scattering (*Soleimanzad et al., 2017*; *Wang et al., 2018*). 300 μm of skull overlaying cortex randomizes the directions of propagation (*Figure 1C*), eliminating the increase in intensity in superficial tissue (*Figure 1D*). By randomizing the propagation angles of photons arriving at the brain surface, skull weights excitation towards the most superficial fluorophore molecules (*Figure 1D*).

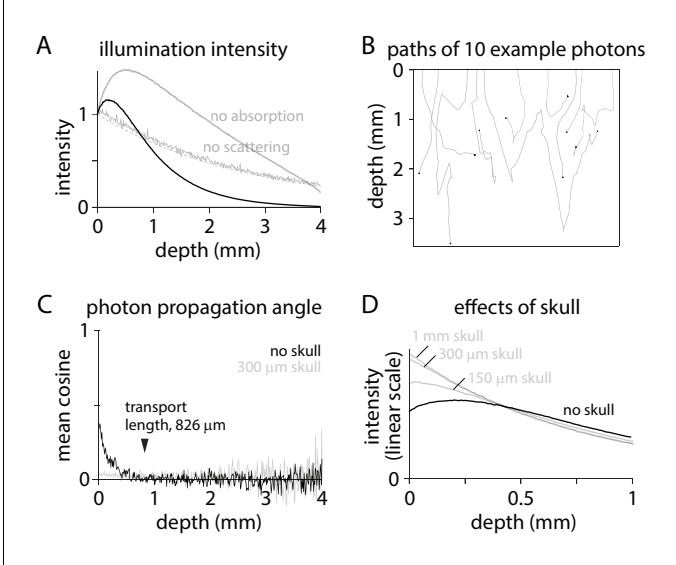

**Figure 1.** Excitation intensity under widefield illumination. (**A**) Intensity as a function of depth, normalized to the intensity at the tissue surface. Dashed line, exponential decay described by the absorption length constant: normalized intensity = exp(- depth * length constant). (**B**) Trajectories in tissue for 10 photons. Tissue surface is at depth = 0. Black circle: location at which each photon was absorbed by the tissue. (**C**) Cosine of the mean propagation angle, relative to the optical axis and perpendicular to the tissue surface. Grey: after 300 µm of skull. (**D**) Intensity in brain tissue without (black) and with skull (grey; 1 mm, 300 µm and 100 µm skull), normalized to the total intensity in brain tissue.

## Fluorescence collection

What percentage of photons from a fluorophore molecule contribute to the image and how does this percentage change with the depth of the source? I simulated fluorescence at 560 nm, in the green-yellow spectrum, from point sources at different depths (*Figure 2A*). Photons that exit the tissue within the 5.7˚ maximum collection angle of the objective are collected and contribute to image formation. 0.58% of photons from a surface source are collected (*Figure 2B*). The collection percentage increases slightly with source depth to ~400 µm, due to scattering, and decreases thereafter (*Figure 2B*). Photons from deep within cortex can contribute as much to detected fluorescence as photons from superficial layers, with collection percentages for sources on the surface and 1 mm deep being equal.

Collected photons are scattered en route to the tissue surface. Photons from a point source in the tissue form a patch of fluorescence on the surface (*Figure 2A*). The diameter of the patch is greater for sources at depth than near the surface. The diameter containing 50% of photons is 10 µm for a source on the brain surface and 860 µm for a 1 mm deep source; the diameter containing 95% of photons is 2.6 mm for a source on the brain surface and 3.9 mm for a 1 mm deep source (*Figure 2C*). Nearby sources, even in the superficial layers of cortex, produce overlapping surface distributions.

The model accurately predicted the surface distribution of photons. In three experiments, a bolus ~50–100 µm in diameter of 0.1 µm fluorescent beads was injected into mouse cortex at depths of 165, 190, and 310 µm. The diameter and depth of the distribution were measured with 2-photon fluorescence microscopy (*Figure 2D*). The diameter of the surface distribution, measured with widefield fluorescence, was ~1 mm (*Figure 2D,E*). The model slightly overestimated the spread toward the edges, but accurately predicted the distribution of the majority of photons (*Figure 2E,F*).

## Effects of focal plane depth, numerical aperture, field of view, and skull

In *Figures 1* and *2*, illumination and fluorescence collection extend through all layers of cortex, peaking in the middle layers. Do illumination and collection change with system optics?

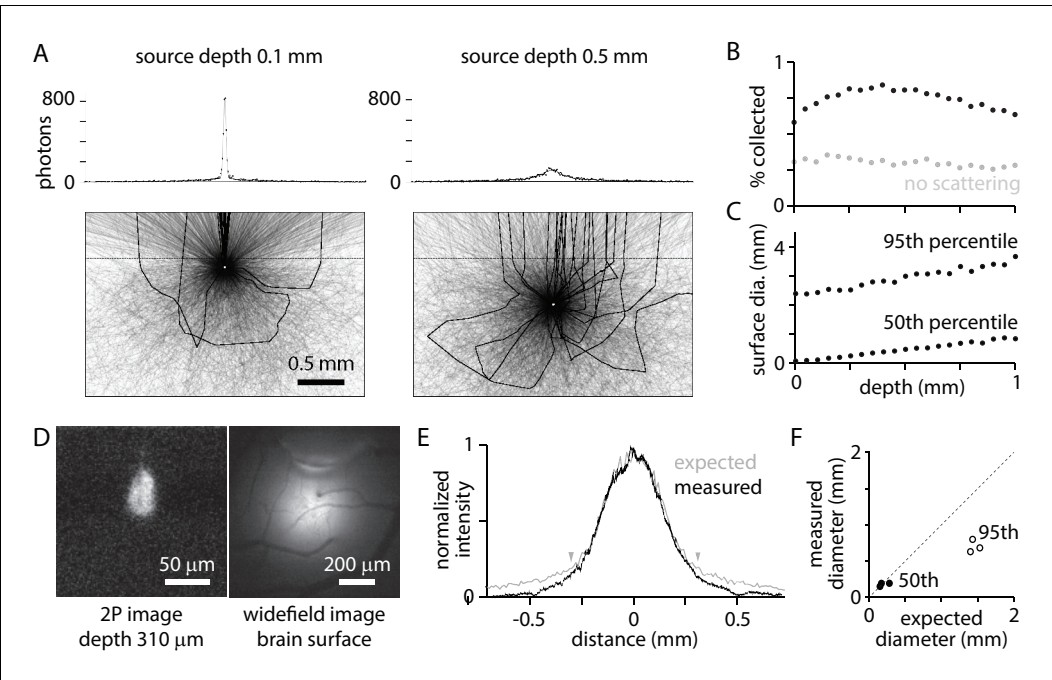

**Figure 2.** Widefield fluorescence collection. (**A**) Trajectories of 3000 photons from a point source (white circle) at 0.1 mm (left) and 0.5 mm (right) below the tissue surface. Black: trajectories of collected photons (23 photons from 0.1 mm, 22 photons from 0.5 mm). Grey: trajectories of photons absorbed in tissue or that exit tissue outside the collection angle of the objective. Dashed line, tissue surface. Histograms: sum of photons at different surface locations. $10^6$ photons (~5000 collected) in 10 µm bins. (**B**) Percentage of photons collected, as a function of source depth. (**C**) Diameter of the patch of fluorescence at the surface, for sources at different depths. Plot illustrates the diameters that include 50% and 95% of captured photons. (**D**) 2-photon and widefield images of a bolus of fluorescent beads injected into mouse cortex, with the focal planes 310 µm below and at the brain surface, respectively. (**E**) Measured surface distribution (from the example in D) and the expected fluorescence distribution, simulated for a point source at a depth of 310 µm. Arrowheads mark the locations between which 75% of photons are expected at the surface. (**F**) Comparison of measured and expected diameters that include 50% (black) and 95% (grey) of photons at the surface, for three experiments with beads at depths of 165, 190, and 310 µm.

One might naively expect focusing up and down through brain tissue to change the weighting of fluorescence across cortical layers. As illustrated in *Figure 3A*, changing the depth of the focal plane below the tissue surface has no effect on illumination, the percentage of photons collected from different depths, or the distribution of fluorescence at the tissue surface (though, of course, adjusting the focus will change the distribution of photons on the camera chip). The lack of effect of changing the focal plane is consistent with the results of *Tian et al., 2011*, who found that focal plane depth mattered little when imaging large regions of uniform fluorescence, and also considered the effects of focal plane on the distribution of photons on the camera chip.

Similarly, one might expect illumination and fluorescence collection to change with the numerical aperture and field of view of the microscope. I compared results with numerical apertures of 0.1 and 0.5. As expected, greater numerical aperture increases collection efficiency substantially, from a maximum collection efficiency of ~0.716% (*Figure 3B*). However, numerical aperture has negligible effect on illumination, weighting of collection with tissue depth or the surface distribution of fluorescence (*Figure 3B*). Peak illumination intensity and collection efficiency remains at ~0.3–0.4 mm below the tissue surface, in superficial layers of cortex, across a range of numerical apertures. The lack of effect of NA is consistent with published results (*Tian et al., 2011*).

Reducing the field of view from 11 to 4.4 mm (corresponding to an increase in objective magnification from x2 to x10; TL2X-SAP and TL10X-2P objectives) has no effect on illumination (*Figure 3C*) but reduces collection efficiency and the diameter of surface distribution of fluorescence

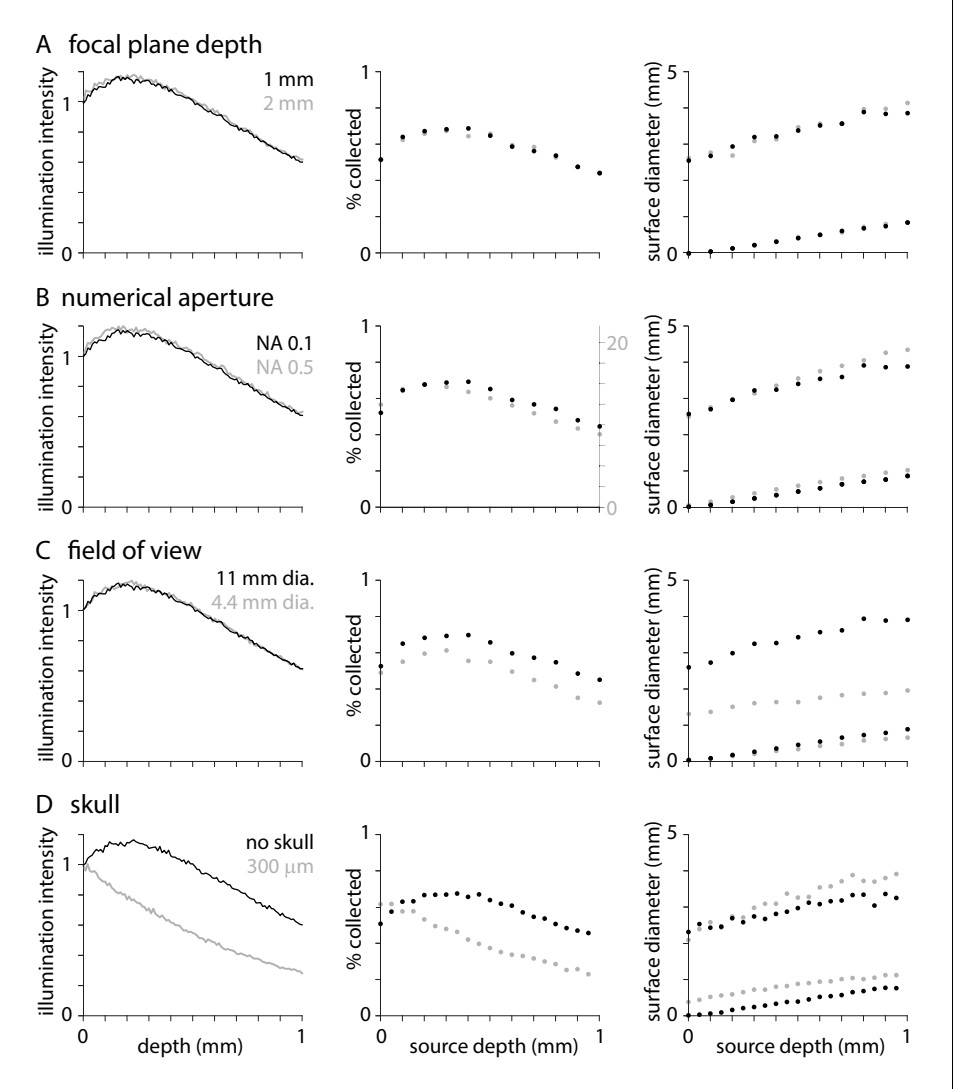

**Figure 3.** Effects of optical parameters on illumination and fluorescence collection. Normalized illumination intensity (c.f. *Figure 1A*), percentage of photons collected (c.f. *Figure 2B*) and diameters that include 50 and 95% of surface fluorescence (c.f. *Figure 2C*) with the objective focused at depths of 1 and 2 mm below the tissue surface (**A**), with objectives of numerical aperture 0.1 and 0.5 (**B**), with objectives of 11 and 4.4 mm field of view (**C**), and when imaging through no and 300 µm of skull. In each plot, black symbols represent results with an objective with NA 0.1 and 11 mm field of view focused 1 mm below the tissue surface with no overlying skull. For collection efficiency, results from different numerical apertures are displayed on different y-axes to facilitate comparison.

(*Figure 3C*). These results illustrate one consequence of the large surface distribution of fluorescence: even a microscope objective with a fairly large field of view can cause vignetting.

Finally, I examined the consequences of imaging through 300 µm of skull. Skull weights illumination and fluorescence collection towards deeper layers of cortex (*Figure 1D*, *Figure 3D*) and slightly broadens the central peak of the surface fluorescence distribution (*Figure 3D*).

In summary, optical parameters typically have modest effects on widefield fluorescence. That said, changing to a higher magnification objective will generally result in a greater numerical aperture and a smaller field of view and together these changes can increase collection efficiency and limit the effective point spread function, albeit by vignetting.

## Fluorescence from tissue under a blood vessel

The absorption and scattering coefficients used in the model were measured in vivo and therefore account for the effects of endogenous molecules such as hemoglobin, with the absorption coefficient in vivo being ~5 times greater than that measured in vitro, largely because of absorption by hemoglobin (*Johansson, 2010*). Nonetheless, in the brain there will be local variations in absorption and scattering, such as in the vicinity of blood vessels, resulting in local effects that are not reflected in *Figures 1–3*.

To obtain a sense of the likely magnitude and spatial scale of local variations in illumination and fluorescence collection, I simulated the effects of surface blood vessels. Blood vessels with circular cross-sections of radius 100 and 250 µm were simulated as regions in which all photons were absorbed. From the Beer-Lambert law, 500 µm of blood transmits <<1% of incident light but 200 µm of blood transmits ~6% ($2.2 \times 10^{-3}$ mol/L hemoglobin, 50% hemoglobin oxygenation, hemoglobin molar extinction coefficient 27,895 cm$^{-1}$M$^{-1}$; *Valley et al., 2020*). Likely this simple simulation slightly overestimates the effects of vessels, particularly small vessels.

Illumination intensity is reduced immediately below a vessel and throughout the deeper cortical tissue, but even tissue immediately under the vessel received substantial illumination (*Figure 4A–C*). This result is expected given the random average direction of travel of photons in deep layers (*Figure 1C*). Similarly, fluorescence collection from under the vessel was reduced but remained substantial. For a large vessel of radius 250 µm, the combined effects on illumination and collection reduce fluorescence to ~5% at 500 µm depth, immediately under the vessel, and to ~45% at 1 mm. Of course, fluorescence is affected less either side of the vessel center line. Clearly even at low numerical apertures (with few oblique angles of illumination at the surface), tissue under vessels can make a substantial contribution to widefield fluorescence.

Vessel diameter and content are dynamic in vivo. Under 2-photon excitation, dilation of a small vessel can cause a substantial decline in fluorescence from an underlying neuron (≥10% ΔF/F from a neuron under a 50 µm diameter vessel during sensory stimulation; *Shen et al., 2012*). Likewise, changes of ≥10% ΔF/F occur in widefield fluorescence near large surface vessels and can be largely separated from changes in tissue fluorescence with appropriate measurements and calculations (*Valley et al., 2020*).

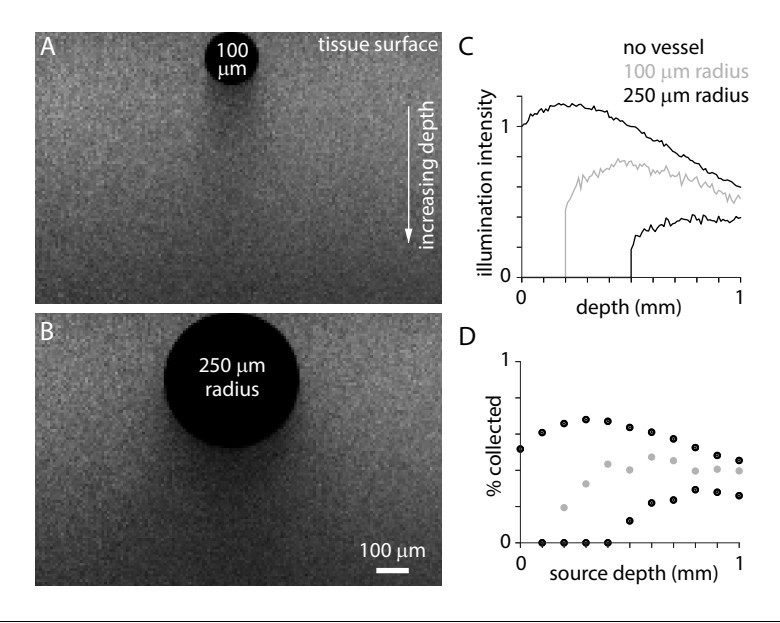

**Figure 4.** Illumination and fluorescence collection under blood vessels. (**A** and **B**) Images illustrating illumination intensity around 100 µm and 250 µm radius surface blood vessels. (**C** and **D**) Illumination intensity (normalized to the brain surface with no blood vessel) and fluorescence collection as a function of depth below the center of the blood vessel.

## Fluorophore expression and the volume from which fluorescence is collected

There are many mouse lines available with fluorophore in sub-populations of neurons, often cells with somata in only one or two layers of cortex. However, the axons and dendrites of most neurons extend into other layers where they may contribute to widefield fluorescence. From which layers do widefield fluorescence signals originate? What percentage of the fluorescence arises from somatic layers?

I measured expression in six mouse lines with GCaMP in laminar sub-populations of excitatory neurons: Slc17a7-Ai93 (all layers), Cux2-Ai93 (layers 2–4), Rorb-Ai93 (layer 4), Rbp4-Ai93 (layer 5), Fezf2-Ai148 (layers 5 and 6), and Ntrs1-Ai148 (layer 6). Fluorescence images of coronal sections from visual cortex were obtained from the Allen Brain Observatory (*Figure 5A*). As expected, there was high expression in layers with GCaMP in somata and moderate expression in other layers, presumably from GCaMP in dendrites and axons (*Figure 5B*).

Multiplying expression, illumination intensity and collection efficiency, each as a function of depth, revealed the relative contributions of different layers to widefield fluorescence (*Figure 5C*). The percentages of fluorescence arising from the somatic layers was high for mice with somatic expression in superficial layers and lower for mice with deep-layer expression (Slc17a7-Ai93 90%, Cux2-Ai93 76%, Rorb-Ai93 26%, Rbp4-Ai93 26%, Fezf2-Ai148 60%, and Ntsr1-Ai148 43%). There are dendrites and axons in all layers so these numbers are an upper bound on the percentage of fluorescence arising from somata.

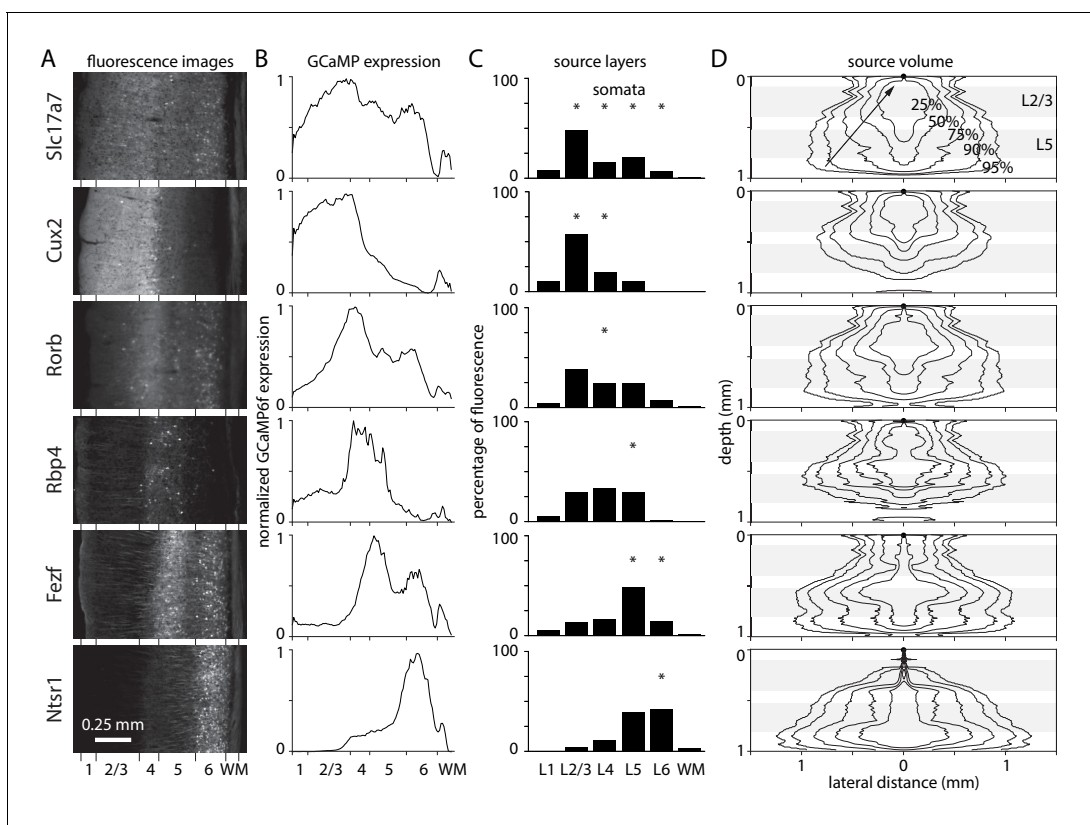

**Figure 5.** Laminar fluorescence in GCaMP mouse lines. (**A**) Fluorescence images from coronal sections of mouse primary visual cortex in six mouse lines, from the Allen Brain Observatory. (**B**) Fluorescence in coronal sections in panel A, as a function of depth and normalized to maximum for each mouse. (**C**) Widefield fluorescence in vivo, the product of expression (panel B), excitation intensity through a cranial window (*Figure 1*), and percentage of fluorescence photons collected (*Figure 2*). Widefield fluorescence is expressed by layer, as a percentage of total fluorescence from pia to the base of white matter. Asterisks: layers with most somata. (**D**) 2-dimensional illustrations of the volume of tissue from which photons propagate to a single location on the brain surface (black circle at 0 depth, 0 lateral distance), arriving within the collection angle of the objective lens. Lines encircle the voxels contributing the most photons, that together contribute 25, 50, 75, 90, and 95% of collected photons. Grey bands indicate the depths of layers 2/3 and 5.

Despite being the surface layer, layer 1 contributes >10% of fluorescence only in Cux2-Ai93 mice (Slc17a7-Ai93 5.8%, Cux2-Ai93 10.2%, Rorb-Ai93 2.8%, Rbp4-Ai93 5.2%, Fezf2-Ai148 3.4%, Ntsr1-Ai148 0.5%). The modest layer 1contribution is expected given that illumination intensity, collection efficiency, and fluorophore expression are all less in layer 1 than layer 2/3 (*Figure 1A*, *Figure 2B*, *Figure 5C*). For all mouse lines, the largest contribution originates from deeper layers (Slc17a7-Ai93 43% from layer 2/3; Cux2-Ai93 57% from layer 2/3; Rorb-Ai93 34% and 27% from layers 2/3 and 4; Rbp4-Ai93 ~ 30% from each of layers 2/3, 4 and 5; Fezf2-Ai148 48% from layer 5; Ntsr1-Ai148 ~ 40% each from layers 5 and 6). However, these numbers are in apparent conflict with the greater correlation between layer 1 (2-photon) and widefield fluorescence than between layer 2/3 and widefield fluorescence observed by *Allen et al., 2017*. The numbers cited above report steady-state fluorescence whereas *Allen et al., 2017* calculated fractional changes in fluorescence (ΔF/F), common practice in the field. Axons and distal dendrites often display larger fractional changes than somata and proximal dendrites (*Helmchen et al., 1996*; *Brenowitz and Regehr, 2007*; *Larkum et al., 2007*; *Xu et al., 2012*) and are major contributors to fluorescence signals in layer 1 and in the neuropil in other layers. How much larger would the fractional change in fluorescence in the neuropil or in layer 1 need to be to substantially change the layer 1 contribution?

Neuropil occupies a larger fraction of the volume of layer 1 than of other layers (neuropil/soma ratio is 0.994 in layer 1 and 0.77–0.84 in deeper layers in albino rat; *Gabbott and Stewart, 1987*) so if the fractional change in fluorescence in the neuropil is increased, layer 1 will contribute a larger percentage of the total change in fluorescence. However, the change is modest. When $\Delta F/F_{neuropil}$ = 10 * $\Delta F/F_{soma}$, layer 1 contributes 6.9% of total fluorescence in Slc17a70Ai93 mice, only ~1% more than when $\Delta F/F_{neuropil}$ = $\Delta F/F_{soma}$.

The effect of increasing the fractional change in layer 1 is greater. If ΔF/F were 10 times greater in layer 1 than in other layers, layer 1 would account for 39% of total fluorescence in Slc17a7-Ai93 mice. However, a 10x difference seems unlikely. While an action potential can evoke cytosolic calcium transients of ~1 μM in cerebellar granule cells axons (*Brenowitz and Regehr, 2007*), even bursts of action potentials evoke more modest changes (~250 nM,≤3 times greater than in the soma) in the distal apical dendrites of layer 2/3 and 5 pyramidal neurons (*Schiller et al., 1995*; *Waters et al., 2003*). The model predicts that layer 1 contributes a minority of the resting fluorescence and likely also of the fractional change in fluorescence in most mouse lines. There are several possible explanations for the observation of *Allen et al., 2017* that the widefield fluorescence correlates more closely with layer 1 than layer 2/3 signals: perhaps expression of GCaMP6f in vGluT1-Cre; Ai93 mice is stronger in layer 1 than in layer 2/3; maybe the olfactory go/no-go decision-making task studied in *Allen et al., 2017* drives a very large ΔF/F in axons and dendrites in layer 1; or are deep-layer pyramidal neurons and their distal apical dendrites in layer 1 are more active than layer 2/3 neurons in this behavioral task?

Fluorescence photons propagating to each location on the brain surface originate from a large volume of underlying tissue, a result of scattering of fluorescence photons en route to the brain surface (*Figure 5D*). In Fezf2-Ai148 and Ntsr1-Ai148 mice, 95% of collected photons arriving at each 10 μm surface pixel are from 1.97 mm³ of underlying tissue, including the full 1 mm depth of cortex and a radius of >1 mm in deep layers. The most compact source volume was 1.08 mm³ in Cux2-Ai93 mice, with photons from all layers and a radius of 830 μm in layer 4. Clearly, nearby surface pixels sample fluorescence from overlapping volumes of tissue. Only pixels >~ 1 mm apart report fluorescence from non-overlapping volumes of tissue. The extremely large volume of contributing tissue is remarkable and underlines the sensitivity of widefield fluorescence imaging to light scattering by brain tissue. The extreme spread of fluorescence from even superficial sources suggests caution when localizing active regions of cortex using widefield fluorescence.

## Discussion

The random-walk model provided several quantitative estimates that assist in the interpretation of widefield fluorescence measurements from brain tissue. Firstly, illumination intensity and fluorescence collection do not decay monotonically from the tissue surface when imaging through a cranial window. Each peaks at a depth of ~3–400 um, in layer 2/3 in mouse cortex, and layer 1 contributes <10% of fluorescence in most mouse lines. Interestingly, although the presence of skull overlying cortex has little effect on resolution, expressed as the surface distribution of photons, skull shifts

illumination and collection towards a monotonic decay, thereby weighting widefield measurements more towards fluorescence from layer 1. Finally, a volume of ~1–2 mm$^3$ of tissue contributes to fluorescence, resulting in resolution on the millimeter scale. Photons from a point source are commonly spread across >1 mm of the cortical surface and the surface distribution can exceed 2 mm in diameter in mice with deep expression. In mice with layer-enriched indicator expression, the main source of fluorescence is commonly the somatic layer in mice with expression in layer 2/3 neurons and outside the somatic layer in mice with expression mainly in neurons in layers 4–6.

Can widefield fluorescence provide finer spatial precision than ~1 mm? The Monte Carlo simulation indicates that there is no simple way to adapt illumination or collection to provide fine spatial precision. Using a higher magnification objective with a field of view of only several millimeters may reduce the measured surface spread but operates by vignetting the image and may thereby shift the apparent location of active tissue. Illuminating a small region of the brain surface will not restrict the excitation volume, much as a laser spot on the brain surface activates opsins > 1 mm away (*Guo et al., 2014*). Spatial precision could be obtained by limiting expression of fluorescent indicator to <1 mm of tissue, with virus for example, but would sacrifice the ability to monitor activity across much of cortex, a key application of widefield imaging.

The Monte Carlo model does not explicitly simulate the temporal dynamics of widefield fluorescence, but temporal information can help locate active tissue. The expansion of activity from a small initial region of tissue can be monitored with fast indicators, such as voltage indicators, enabling the center of the initiation site to be located with fine spatial precision (*Petersen et al., 2003*; *Mohajerani et al., 2013*). In addition, widefield fluorescence can locate with high spatial precision sites at which waves of activity converge. For example, the border between visual areas can be located to within tens of micrometers with GCaMP6s and a stimulus that drives converging waves of activity either side of the border (*Zhuang et al., 2017*). In contrast, the model indicates that locating a border from activity on only one side of the border is extremely imprecise, on the order of a millimeter.

In summary, widefield fluorescence at the brain surface is a weighted sum of photons from fluorophores distributed through 1–2 mm$^3$ of underlying cortical tissue. Illumination intensity and collection efficiency peak not at the tissue surface, but in layer 2/3. Most fluorescence arises from the somatic layer in mice with expression in layer 2/3 neurons and from outside the somatic layer in mice with expression mainly in neurons in layers 4–6. The contributing volume extends laterally, by a radius of ~1 mm, several times larger than the width of a cortical column and comparable to the diameters of some cortical areas (~0.5–3 mm in mice). Widefield fluorescence measured at most surface locations is a weighted average across multiple cortical columns and often more than one cortical area.

## Materials and methods

### Monte carlo model

Photon trajectories, light intensities and tissue heating were calculated using a Monte Carlo random-walk model implemented in Python. The model was almost identical to that in several previous studies (*Wang et al., 1995*; *Stujenske et al., 2015*; *Podgorski and Ranganathan, 2016*; *Wang et al., 2020*). Individual photons or packets of photons moved stochastically through the 3-dimensional volume, in which they were subjected to absorption and scattering by the tissue. Scattering angles relative to the optical axis were calculated with the Henyey-Greenstein phase function.

For most simulations, the radius of the volume was 8 mm and the tissue depth 4 mm. The cranial window was modelled as a 7.5 mm glass coverslip, surrounded by skull. Intact skull was modeled as a 150 µm-thick layer of bone. The tissue surface was planar and in contact with the glass coverslip and skull. Each voxel of the model was 10 × 10 × 10 µm.

Absorption and scattering coefficients and anisotropy parameters were from the literature. Absorption and scattering coefficients used here were measured from human grey matter in vivo and the absorption coefficient is ~5 times greater than that measured in vitro, likely because of absorption by blood (*Johansson, 2010*). Values for grey matter were from *Johansson, 2010*, for skull from *Firbank et al., 1993* and *Ugryumova et al., 2004*. For 480 nm illumination, absorption and scattering coefficients and anisotropy were: grey matter 0.37 mm$^{-1}$, 11 mm$^{-1}$ and 0.89; skull

0.12 mm$^{-1}$, 35 mm$^{-1}$ and 0.9. For 560 nm fluorescence photons in grey matter, absorption coefficient 0.26 mm$^{-1}$, scattering coefficient 10 mm$^{-1}$, and anisotropy 0.89. A recent study concluded that the scattering was greater in mouse cortical slices, with scattering coefficient 21.1 mm$^{-1}$ at 473 nm (*Yona et al., 2016*). Using this increased scattering coefficient here resulted in similar results, but with maximum illumination intensity through a cranial window at a more superficial location, 130 µm below the tissue surface.

From a scattering coefficient of 11 mm$^{-1}$, transport length in mouse cortical grey matter at 480 nm is (1/11) / (1–0.89)=826 µm. Transport length is the distance over which the direction of propagation of photons is randomized as a result of scattering.

Mesoscale widefield imaging is typically performed through a low-magnification, long working distance, dry, low numerical aperture (NA) objective, often a dissecting microscope lens or camera lens. Here I simulated one such objective, TL2X-SAP from Thorlabs, for which optical parameters are readily available: magnification 2, numerical aperture 0.1, working distance 56.3 mm, effective focal length 100 mm, field number 22. The optical axis of the objective was perpendicular to the tissue surface. Photons exiting the objective and arriving at the tissue surface were tilted at a maximum angle of 5.74° (NA = n sinθ, n = 1) to the optical axis.

Simulations were also performed with the numerical aperture and field of view of a higher magnification objective from the same series, TL10X-2P. For TL10X-2P, magnification 10, numerical aperture 0.5, working distance 7.7 mm, effective focal length 20 mm, field number 22.

Collection was simulated for photons from sources near the center of the field of view. Towards the periphery of the field of view, photons at the steepest collection angles can pass outside the radius of the front window of the objective, reducing collection efficiency. The decline in collection for off-center photons will depend on the objective. The Thorlabs x2 objective has a 7 mm radius front window and 56.3 mm working distance so collection efficiency will be reduced for sources 56.3 * tan(5.7°) = ~1.5 mm from the center of the field of view. Dissecting microscope and camera lenses, both often used in widefield microscopes, typically have larger front windows, some tens of millimeters in radius. Such super-large diameter objectives can offer more consistent collection over fields of view in excess of several millimeters.

Monte Carlo code, including figures, is available at https://doi.org/10.6084/m9.figshare.12317414.v1.

## Measured surface distributions

0.1 µm diameter fluorescent polystyrene beads (Molecular probes) were injected into the cortex of an anesthetized adult mouse (2% isoflurane) with a cranial window sealed with a 5 mm diameter coverglass. Injection was performed through a glass patch pipette with the tip bumped to prevent blockage, inserted into the brain under 2-photon visual guidance through a ~ 0.5–1 mm hole drilled in the coverglass. Beads were injected with positive pressure. 2-photon and widefield images were acquired through a Nikon x16/NA0.8 objective. The depth of the center of the injection was determined from a 2-photon z-stack and the corresponding expected distribution calculated using the model, assuming a point source at the appropriate depth and a TL10X-2P objective.

Animal experiments were performed in accordance with the recommendations in the *Guide for the Care and Use of Laboratory Animals* of the National Institutes of Health. All animals were handled according to Allen Institute for Brain Science institutional animal care and use committee protocol 1806.

## Expression patterns

GCaMP6f expression was measured in six mouse lines: Slc17a7-IRES-Cre;Camk2a-tTA;Ai93(TITL-GCaMP6f), Cux2-CreERT2;Camk2a-tTA;Ai93(TITL-GCaMP6f), Rorb-IRES2-Cre;Camk2a-tTA;Ai93(TITL-GCaMP6f), Rbp4-Cre_KL100;Camk2a-tTA;Ai93(TITL-GCaMP6f), Fezf2-CreER2;Ai148(TIT2L-GC6f-ICL-tTA2), and Ntrs1-Cre_GN220; Ai148(TIT2L-GC6f-ICL-tTA2). Abbreviated names: Slc17a7-Ai93, Cux2-Ai93, Rorb-Ai93, Rbp4-Ai93, Fezf2-Ai148, and Ntrs1-Ai148.

Fluorescence in 2-photon images of fixed, coronal sections was used to estimate GCaMP expression. Images were obtained from the Allen Institute (http://observatory.brain-map.org/visualcoding/transgenic). For each mouse, an image was selected of primary visual cortex between the AM/PM and LM/AL borders, at ~2.7 mm posterior from bregma. Images were aligned manually and rotated

to place the brain surface parallel to one edge of the image. To correct inter-mouse variability, images were scaled such that the brain surface and layer 6-white matter border were at 0 and 1 mm depth. To estimate expression, fluorescence was summed over a 0.75 mm-wide strip of cortex, perpendicular to the brain surface.

## Acknowledgements

I thank the Allen Institute founder, Paul G Allen, for his vision, encouragement, and support. I thank Kevin Takasaki, Doug Ollerenshaw, Jun Zhuang and Natalia Mesa for comments on the manuscript, and Soumya Chatterjee, Bryan MacLennan and Natalia Mesa for assistance with imaging experiments.

## Additional information

### Funding

| Funder | Author |
| --- | --- |
| Allen Institute for Brain Science | Jack Waters |

The funders had no role in study design, data collection and interpretation, or the decision to submit the work for publication.

### Author contributions

Jack Waters, Conceptualization, Software, Formal analysis, Funding acquisition, Validation, Visualization, Methodology, Writing - original draft, Project administration, Writing - review and editing

### Author ORCIDs

Jack Waters (iD) https://orcid.org/0000-0002-2312-4183

### Decision letter and Author response

Decision letter https://doi.org/10.7554/eLife.59841.sa1
Author response https://doi.org/10.7554/eLife.59841.sa2

## Additional files

### Supplementary files

• Transparent reporting form

### Data availability

Code and results are publicly available on figshare. https://doi.org/10.6084/m9.figshare.12317414.v1.

The following dataset was generated:

| Author(s) | Year | Dataset title | Dataset URL | Database and Identifier |
| --- | --- | --- | --- | --- |
| Waters J | 2020 | Sources of widefield fluorescence from the brain. | https://figshare.com/articles/widefield_fluorescence_ipynb/12317414/1 | figshare, 10.6084/m9.figshare.12317414.v1 |

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
