## [Decision Letter]

**Acceptance summary:**

This manuscript characterizes the source of fluorescence observed in typical widefield imaging experiments using a Monte Carlo simulation. The simulation includes the propagation of photons in the brain. The model was further validated experimentally by beads injected into the mouse brain and transgenic mice expressing GCaMP in a different subset of neurons. The main conclusion is that widefield fluorescence at the brain surface is a weighted sum of photons from fluorophores distributed throughout a large area (millimeters) of cortical tissue. This work will be useful for the community when interpreting fluorescence widefield imaging in the brain.

**Decision letter after peer review:**

Thank you for submitting your article "Sources of widefield fluorescence from the brain" for consideration by *eLife*. Your article has been reviewed by three peer reviewers, and the evaluation has been overseen by a Reviewing Editor and John Huguenard as the Senior Editor. The reviewers have opted to remain anonymous.

The reviewers have discussed the reviews with one another and the Reviewing Editor has drafted this decision to help you prepare a revised submission.

Your manuscript characterizes the source of fluorescence observed in typical widefield imaging experiment uses a Monte Carlo model to simulate the propagation of photons in the brain. These simulations include several lines of transgenic mice, that exhibit different laminar expression patterns. The main conclusions is that widefield fluorescence at the brain surface is a weighted sum of photons from fluorophores distributed throughout a large volume of underlying cortical tissue and only pixels more than 1 mm apart report fluorescence from non-overlapping volumes of tissue.

While reviewers agree that the manuscript includes important information about widefield imaging for the neuroscience community, they also find several concerns that can be addressed by additional analyses (see below). In particular, there was a significant concern about the lack of experimental validation of the simulation. We suggest that you find imaging data of point sources or sparse neurons in the cortex in literature and use them for validation of the simulation results. In case such data is not available, please revise the title and the last sentence of Abstract to include "by Monte Carlo simulation", and clarify the limitations of the current simulation and how this work fits within the field.

Reviewer #1:

Widefield fluorescence microscopy is increasingly used in neuroscience research to monitor the activity of large populations of neurons across the brain, but the spatial extent of signals detected by this technique is not well understood. This manuscript simulated light propagation through brain using a Monto Carlo model of photon interaction with tissue, which is similar to previous published models and incorporated several key parameters from the literature. Conducting this simulation under the constrain of the typical imaging conditions of widefield fluorescence microscopy, the author calculated how illumination intensity and collection efficiency will change as a function of depth. Furthermore, the author multiplied these model-derived coefficients with GCaMP expression levels in several layer-selective mouse lines evaluated from coronal tissue sections by Allen Brain Observatory, and obtained quantitative estimations of the spatial extent of signals that would contribute to widefield imaging in these lines. The main conclusions appear to be that "widefield fluorescence at the brain surface is a weighted sum of photons from fluorophores distributed throughout a large volume of underlying cortical tissue" and "only pixels more than 1 mm apart report fluorescence from non-overlapping volumes of tissue".

In general, the imaging field will welcome a quantitative estimation of the spatial extent of collected signals using simulation models as described in this manuscript. Such quantitative information will help interpret the experimental data collected by wide-field microscopy in related mouse lines, particularly regarding the spatial organization and regional boundary of cortical activity patterns. The modeling and estimation steps in this manuscript are well laid out and justified. The simulation outcomes, some are not obvious, are also explained with an in-depth consideration of imaging conditions and biological factors. There are a couple of remaining concerns about the significance and strength of the author's conclusions.

First, the general qualitative conclusions of the paper, such as "widefield fluorescence at the brain surface is a weighted sum of photons from fluorophores distributed throughout a large volume of underlying cortical tissue", and "Most fluorescence arises from the somatic layer in mice with expression in layer 2/3 neurons and from outside the somatic layer in mice with expression mainly in neurons in layers 4-6", are mostly in line with the general understanding and expectation in the field. The impact statement of the author, "Deep layer cortical neurons are often a major source of fluorescence in widefield imaging experiments from transgenic mouse lines", also doesn't appear to add much new insight. I'm wondering whether some of the less obvious simulation results, such as the non-monotonic changes of illumination intensity (Figure 1) and collection efficiency (Figure 2), will add some new predictions for wide-field microscopy.

Second, the quantitative estimates, such as the 1 mm lateral extent of signal spread, are purely based on simulation. As this is an important parameter relatable to the scale of cortical columns and area size, it would be ideal to obtain some experimental validation. For example, maybe one can take advantage of potentially existing wide-field imaging data where a small amount of virus is used to express GCaMP in a local volume, which is an experimental design touched upon by the author in the Discussion.

Reviewer #2:

The author evaluates using Monte Carlo simulation of the excitation and emission of photons from functional imaging indicators, such as GCAMP. There is also data from leading mouse lines but the use of mostly to model different GCAMP distributions (based on actually expression) rather than an experimental confirmation. The author primarily concerns himself with the occurrence of wide-field single photon fluorescence events. While there has been some theoretical work on scattering in the past, this manuscript combines simulation results with density measurements using mouse lines that target expression to various layers. The work helps to address long-standing questions in the field concerning the relative laminar contribution and resolution of wide field functional imaging experiments.

While the paper is notable for addressing an important question, it makes heavy use of Monte Carlo simulation. Such results lack important ground truth conclusions. Nonetheless, the theoretical treatment of scattered photons combined with some experimental data on expression is a significant contribution. The paper is also relatively easy to read.

Overall the paper does place important limits on the resolution of wide field microscopy, while the effective PSF is large for fluorescence originating at deep sites, the author does articulate that the center of activation and borders (when comprised of travelling activation fronts) can contain much higher resolution information.

The Abstract and title do not really conclude much, they could be more definitive about actual findings

Introduction.

In the first paragraph, the author should say fluorescent photons propagate from the source randomly, not necessarily towards the brain surface.

Figure 1. The figure could use better labelling, so that one does not have to actively consult the Legend.

Figure 1D for the access labelled intensity is assumed that this is a linear scale, but this should be indicated more directly.

Figure 2 indicates the fate of wide-field fluorescence at sources of different depths. Surprisingly, the percentage of collected photon is relatively similar between 0 and 1mm depth. What changes is the radius of emitted fluorescence at the surface. This to me is clearly demonstrated by the Monte Carlo simulation, but it may be appropriate to backup these measurements with some empirical findings. One could imagine embedding beads or other fluorescent objects at different depths in brain tissue (or best a live mouse) and examining collected fluorescence (a fluorescent pipette?).

Figure 3 Using data from various Allen Institute mouse lines, Waters, again, simulated photon collection at the surface for different scenarios. In all lines, even with relatively restricted expression, the contribution of photons at the surface was from a very wide area. These detected photons in all mouse lines could be from sources as far apart as 1mm. While there was data used from actual mice (distribution), the paper does not contain empirical measurements. Here, again, this is plugging-in density values and making estimates using the Monte Carlo model. I would feel better about these strong conclusions if there could be some actual measurements from discrete fluorescent sources. A couple of data points would go a long way to convincing one that these models take everything into consideration: skull, cover slip, age of tissue and white matter content etc. The author could add some empirical measurements to the current manuscript.

The authors should discuss papers that report differences in wide field maps across cortical lamina and whether these are still expected given their findings (Allen et al., 2017, already cited but for another point).

Perhaps, the author would like to discuss and expand upon a previous empirical study performed by Silasi et al., 2016, Figure 3B, which measured the point-spread function of sub-resolution beads with and without the bone and overlying coverslip/dental cement preparation. Such work does demonstrate that some experimental measurements would be possible albeit looking at fluorescence coming from a single deep tissue source in vivo would be most informative. Silasi et al., 2016, which is already cited but not for the PSF and bone measurement

Reviewer #3:

General assessment: To characterize the source of fluorescence observed in typical widefield imaging experiments, Waters uses a Monte Carlo model to simulate the propagation of photons in the brain. These simulations include several lines of transgenic mice, that exhibit different laminar expression patterns. One limitation of the current study is that it provides little background to contextualize it within the broader literature on widefield imaging. In addition, there is limited detail about how the model's simplifying assumptions might influence the predicted spatial distribution of the fluorescence signal. Changes to the text would improve the article, broaden the potential audience and be useful to many readers. With successful modifications, the article could in principle be a candidate for publication in *eLife*.

1) It would be useful to place the current article in the context of other widefield imaging studies. A reader would want to know the limits of previous studies and the reason the current study needed. If space allows, a brief scholarly survey could be provided. For example, the authors could discuss other models that have been used to evaluate the performance of optical imaging methods (such as NAOMi, described in doi: https://doi.org/10.1101/726174), as well as experiments to characterize the source of fluorescence in single photon calcium imaging experiments. For example, Allen et al., 2017 (cited in the current article), performed a comparative 1P and 2P volumetric imaging from the same region which enabled them to determine the relative contributions of layer 1 vs. 2/3 in widefield imaging.

2) It would be worth mentioning how the predictions of the model might be experimentally tested.

3) In the Materials and methods section, the author mentions that their simulations do not take into account the effect of cerebral vasculature. It would be useful to move this topic to a more prominent location in the text and to discuss the potential consequences of this simplifying assumption.

4) The author mentions that the model does not take into account the observation that fractional changes in calcium fluorescence in neuropil are often larger than in the soma. It would be useful to discuss this topic in more detail. How might this omission effect their interpretation of sources of fluorescence changes in widefield imaging experiments? Is it possible to estimate the magnitude of this effect or to include this feature in the current or future simulations? Could layer 1, which is primarily axons and dendrites, potentially contribute a great degree to fluorescence changes in 1P imaging than layer 2/3?

---

## [Author Response]

While reviewers agree that the manuscript includes important information about widefield imaging for the neuroscience community, they also find several concerns that can be addressed by additional analyses (see below). In particular, there was a significant concern about the lack of experimental validation of the simulation. We suggest that you find imaging data of point sources or sparse neurons in the cortex in literature and use them for validation of the simulation results. In case such data is not available, please revise the title and the last sentence of Abstract to include "by Monte Carlo simulation", and clarify the limitations of the current simulation and how this work fits within the field.

I added a data set to test the accuracy of the model, injecting beads into cortex in 3 mice, measuring the distribution and depth of each injection with 2-photon microscopy, and comparing the widefield fluorescence surface distribution with the distribution predicted by the model. The experiment is beset with practical difficulties (for example, the bolus is rarely spherical and there are usually a few beads in the injection tract) and the model isn’t a perfect match (I assumed a point source and similar but not identical objective) so I think it’s a fairly course comparison. Nevertheless, the model accurately predicted the shape and width of the central peak of the surface distribution. The tails of the distribution were not accurately predicted by the model, with the model predicting a larger spread, but the tails contain only ~25% of the photons. I believe the results indicate that the model is sufficiently accurate to support the conclusions of the paper. The comparison is presented in Figure 2.

Reviewer #1:[…] First, the general qualitative conclusions of the paper, such as "widefield fluorescence at the brain surface is a weighted sum of photons from fluorophores distributed throughout a large volume of underlying cortical tissue", and "Most fluorescence arises from the somatic layer in mice with expression in layer 2/3 neurons and from outside the somatic layer in mice with expression mainly in neurons in layers 4-6", are mostly in line with the general understanding and expectation in the field. The impact statement of the author, "Deep layer cortical neurons are often a major source of fluorescence in widefield imaging experiments from transgenic mouse lines", also doesn't appear to add much new insight. I'm wondering whether some of the less obvious simulation results, such as the non-monotonic changes of illumination intensity (Figure 1) and collection efficiency (Figure 2), will add some new predictions for wide-field microscopy.

I have rewritten the impact statement and Abstract to emphasize new insights.

Second, the quantitative estimates, such as the 1 mm lateral extent of signal spread, are purely based on simulation. As this is an important parameter relatable to the scale of cortical columns and area size, it would be ideal to obtain some experimental validation. For example, maybe one can take advantage of potentially existing wide-field imaging data where a small amount of virus is used to express GCaMP in a local volume, which is an experimental design touched upon by the author in the Discussion.

I’ve obtained validation results from 3 mice, discussed above and added to Figure 2. The new results support the prediction, made by the model, that signal spreads over 1 mm or more laterally.

Reviewer #2:[…] The Abstract and title do not really conclude much, they could be more definitive about actual findings

I’ve rewritten the Abstract to summarize more results.

Introduction.In the first paragraph, the author should say fluorescent photons propagate from the source randomly, not necessarily towards the brain surface.

I’ve rewritten this statement.

Figure 1. The figure could use better labelling, so that one does not have to actively consult the Legend.

I’ve added panel titles and labels.

Figure 1D for the access labelled intensity is assumed that this is a linear scale, but this should be indicated more directly.

I’ve added ‘linear scale’ to the axis title.

Figure 2 indicates the fate of wide-field fluorescence at sources of different depths. Surprisingly, the percentage of collected photon is relatively similar between 0 and 1mm depth. What changes is the radius of emitted fluorescence at the surface. This to me is clearly demonstrated by the Monte Carlo simulation, but it may be appropriate to backup these measurements with some empirical findings. One could imagine embedding beads or other fluorescent objects at different depths in brain tissue (or best a live mouse) and examining collected fluorescence (a fluorescent pipette?).

I followed the reviewer’s suggestion, embedding beads in a live mouse. During experiments, I was unable to visualize the pipette tip deeper than ~400 μm and was therefore restricted to measurements from superficial tissue. The results indicate that the model accurately predicts the effects of tissue for superficial fluorescence sources. Though I cannot provide direct experimental support, it is likely that the model is also accurate for deeper sources.

Figure 3 Using data from various Allen Institute mouse lines, Waters, again, simulated photon collection at the surface for different scenarios. In all lines, even with relatively restricted expression, the contribution of photons at the surface was from a very wide area. These detected photons in all mouse lines could be from sources as far apart as 1mm. While there was data used from actual mice (distribution), the paper does not contain empirical measurements. Here, again, this is plugging-in density values and making estimates using the Monte Carlo model. I would feel better about these strong conclusions if there could be some actual measurements from discrete fluorescent sources. A couple of data points would go a long way to convincing one that these models take everything into consideration: skull, cover slip, age of tissue and white matter content etc. The author could add some empirical measurements to the current manuscript.

I added three measurements of surface fluorescence in a live mouse, imaged through a coverslip. As discussed above, the measured and predicted (by the model) distributions were reasonably well matched. These results provide additional confidence that the model takes all the relevant parameters into account, including coverslip, age of tissue, white matter content. I did not attempt to measure the scattering effects of skull, but most of the paper concerns simulations of widefield fluorescence through a cranial window.

The authors should discuss papers that report differences in wide field maps across cortical lamina and whether these are still expected given their findings (Allen et al., 2017, already cited but for another point).

I have added new simulations and associated discussions on laminar differences, including the observations of Allen et al. These topics are addressed in the fourth paragraph of the Results subsection “Fluorophore expression and the volume from which fluorescence is collected”.

Perhaps, the author would like to discuss and expand upon a previous empirical study performed by Silasi et al., 2016, Figure 3B, which measured the point-spread function of sub-resolution beads with and without the bone and overlying coverslip/dental cement preparation. Such work does demonstrate that some experimental measurements would be possible albeit looking at fluorescence coming from a single deep tissue source in vivo would be most informative. Silasi et al., 2016, which is already cited but not for the PSF and bone measurement

I have added new results measuring lateral spread of signal from beads. Unlike Silasi et al., 2016, the new measurements were through a cranial window, not through skull.

Silasi et al., 2016, reported fluorescence 48.6 ± 5.7 µm in diameter with an unknown number and spread of 10 µm beads on the brain surface. Without knowing the spread at the brain surface, it’s impossible to determine the effect of skull on lateral resolution. Furthermore, Silasi et al., 2016, do not indicate whether the diameter measurement was FWHM or some other measure of diameter. In short, it’s hard to interpret the Silasi results, limiting my ability to compare and discuss.

Reviewer #3:1) It would be useful to place the current article in the context of other widefield imaging studies. A reader would want to know the limits of previous studies and the reason the current study needed. If space allows, a brief scholarly survey could be provided. For example, the authors could discuss other models that have been used to evaluate the performance of optical imaging methods (such as NAOMi, described in doi: https://doi.org/10.1101/726174), as well as experiments to characterize the source of fluorescence in single photon calcium imaging experiments. For example, Allen et al., 2017 (cited in the current article), performed a comparative 1P and 2P volumetric imaging from the same region which enabled them to determine the relative contributions of layer 1 vs. 2/3 in widefield imaging.

I have expanded the Introduction to provide context, including a brief summary of previous modeling studies. I also ran additional simulations to provide insight into the relative contribution of layer 1, discussed further under point 4, below.

2) It would be worth mentioning how the predictions of the model might be experimentally tested.

The revised paper includes a small data set that tests some of the key predictions.

3) In the Materials and methods section, the author mentions that their simulations do not take into account the effect of cerebral vasculature. It would be useful to move this topic to a more prominent location in the text and to discuss the potential consequences of this simplifying assumption.

The literature includes estimates of the effects of vasculature on 2-photon fluorescence from underlying neurons, but no estimates for widefield fluorescence, to my knowledge. For the revised manuscript I implemented a simple model of vasculature, resulting in a new section in the Results (subsection “Fluorescence from tissue under a blood vessel”) and a new figure (Figure 4). That widefield fluorescence can arise from under the vessel is hardly surprising and the results are really only a first guess given the exceedingly simple nature of the vascular model, but the simulation provides quantitative estimates of the effects of surface vasculature on widefield fluorescence and adding results on the effects of vasculature places the topic in a more prominent location.

4) The author mentions that the model does not take into account the observation that fractional changes in calcium fluorescence in neuropil are often larger than in the soma. It would be useful to discuss this topic in more detail. How might this omission effect their interpretation of sources of fluorescence changes in widefield imaging experiments? Is it possible to estimate the magnitude of this effect or to include this feature in the current or future simulations? Could layer 1, which is primarily axons and dendrites, potentially contribute a great degree to fluorescence changes in 1P imaging than layer 2/3?

I have run additional simulations with modified neuropil and layer 1 signals, addressing this topic at some length in the revised Results section (subsection “Fluorophore expression and the volume from which fluorescence is collected”, fourth paragraph).